# Effects of pay rate and instructions on attrition in crowdsourcing research

Carolyn M. Ritchey[1]*, Corina Jimenez-Gomez[2], Christopher A. Podlesnik[2]

**1** Department of Psychological Sciences, Auburn University, Auburn, AL, United States of America,
**2** Department of Psychology, University of Florida, Gainesville, FL, United States of America

\* cmr0112@auburn.edu

## Abstract

Researchers in social sciences increasingly rely on crowdsourcing marketplaces such as Amazon Mechanical Turk (MTurk) and Prolific to facilitate rapid, low-cost data collection from large samples. However, crowdsourcing suffers from high attrition, threatening the validity of crowdsourced studies. Separate studies have demonstrated that (1) higher pay rates and (2) additional instructions–i.e., informing participants about task requirements, asking for personal information, and describing the negative impact of attrition on research quality–can reduce attrition rates with MTurk participants. The present study extended research on these possible remedies for attrition to Prolific, another crowdsourcing market-place with strict requirements for participant pay. We randomly assigned 225 participants to one of four groups. Across groups, we evaluated effects of pay rates commensurate with or double the US minimum wage, expanding the upper range of this independent variable; two groups also received additional instructions. Higher pay reduced attrition and correlated with more accurate performance on experimental tasks but we observed no effect of additional instructions. Overall, our findings suggest that effects of increased pay on attrition generalize to higher minimum pay rates and across crowdsourcing platforms. In contrast, effects of additional instructions might not generalize across task durations, task types, or crowdsourcing platforms.

## Introduction

Over the last decade, crowdsourcing marketplaces such as Amazon Mechanical Turk (MTurk) and Prolific have provided participants for up to half of published studies in psychology and other social sciences [1]. MTurk is by far the most popular recruitment platform to offer rapid data collection from large samples at a relatively low cost and increased diversity of samples beyond the typical college undergraduate population. Prolific offers many of the same benefits as MTurk, but is geared toward academic researchers. Of course, these and other crowdsourcing platforms are not entirely distinct, given that there is likely some overlap in participant populations. Moreover, the characteristics of each of these platforms is likely to vary significantly over time. For example, Difallah and colleagues reported that the MTurk participant pool is largely comprised of new individuals every few years, with the half-life of participants ranging between 12 and 18 months [2]. Nevertheless, some crowdsourcing platforms have

**Funding:** The author(s) received no specific funding for this work.

**Competing interests:** The authors have declared that no competing interests exist.

distinct features that might make recruitment more convenient for psychological research. For example, Prolific uniquely offers participant prescreening based on multiple factors, including self-reported clinical diagnoses (e.g., autism, depression).

Despite the many benefits of using crowdsourcing in psychological research, one general limitation of such online research is attrition. Attrition can occur for many reasons–participants might drop out of a study due to an experimental condition (e.g., punishment versus no punishment [3]), exogenous factors such as interruptions (e.g., attending to a child), or technical difficulties. Anonymity in online research also minimizes the social cost of dropping out of a study. Attrition in online experiments is significant not only because of its potential to increase the time and cost of conducting online research, but also because it can jeopardize internal validity when it occurs disproportionally across experimental conditions.

Research suggests that attrition rates are similar across studies conducted using both Prolific and MTurk. For example, Palan and Schitter reported ~25% attrition after two weeks for inexperienced Prolific participants–i.e., participants who had completed only one study on that platform [4]. Kothe & Ling also reported 25% attrition in a Prolific study after one year and ~40% attrition after 19 months [5]. Attrition rates are similarly high in MTurk studies spanning several weeks to months (30%) or up to one year (55%) [6]. However, attrition often is not reported in shorter-term studies with MTurk [7, 8] or Prolific.

A few studies, however, examined possible remedies for attrition in online research. For example, Zhou and Fishbach examined the effects of *additional instructions* with MTurk participants, defined as (1) a prewarning–i.e., informing participants about the task, (2) personalization–i.e., asking for personal information (MTurk ID) on the consent form, and (3) appealing to conscience–i.e., explaining the negative effects of attrition on research quality [8]. The combination of these instructions reduced attrition by more than half in a 5-min survey task compared to no additional instructions (from ~50% to ~20%) [9–11].

Increasing pay is a second empirically validated method for reducing attrition in MTurk studies [7, 12]. For example, Auer et al. evaluated attrition in a two-part survey study, with each part lasting approximately 30 min [12]. Across groups, Auer et al. manipulated both initial pay for completing part 1 –ranging from $0.50 to $2.00 –and pay increase from part 1 to part 2, ranging from 0% to 400%. Both initial pay and pay increase significantly predicted attrition, with attrition ranging from 30% to 82% across the highest- and lowest-paid groups, respectively. In line with these findings, Crump et al. reported in a concept-learning task that attrition was higher in a low-incentive condition (total pay of $0.75; 25% attrition) compared to a high-incentive condition (total pay of up to $4.50; 11% attrition) [7]. This limited research suggests that advertised pay likely is an important factor in minimizing attrition among MTurk participants.

The purpose of the present study was to extend findings on effects of (1) higher pay and (2) additional instructions on attrition to Prolific participants. Given the benefits of some of this platform's features for psychological researchers, we sought only to test these manipulations in novel ways within the Prolific platform. For example, we evaluated effects of pay and instruction manipulations alone and in combination, a novel contribution to this research area. We hypothesized that both higher pay and additional instructions would reduce attrition, with greater reductions in attrition when combining these manipulations.

## Methods

### Participants

The study protocol was approved by the Auburn University Institutional Review Board and all participants provided written consent for participation. A power analysis based on previous research [8, 12] indicated that a sample size of at least 37 participants per group would ensure

adequate power (>.80) to detect an effect of pay with an effect size (f) of 0.23 and instructions with an effect size (w) of 0.57. We recruited a total of 225 participants from www.Prolific.co. Forty participants per group completed operant key-pressing tasks, and we collected demographic information from 152 participants (67.6%) completing the post-experiment survey. Those participants ranged in age from 18 to 63 (M = 31.5, SD = 10.8), and just over half identified as male (54.6%). The majority of those 152 participants (99.3%) resided in the United States, and one did not report this information. Detailed demographic information is included with supplemental materials. Although it is possible to prescreen participants on Prolific–for example, filtering out participants with low approval rates for previous work–we did not apply any prescreening filters. Research suggests that, unlike MTurk, Prolific generally provides high-quality data without prescreening [13].

## Apparatus and procedure

We programmed the experimental tasks using Inquisit [14]. The experiment included two brief (~10 min) probabilistic reversal learning tasks commonly used to examine inhibitory control and behavioral flexibility [15, 16]. These included a two-choice (simultaneous) and successive (go/no-go) task, comprising Part 1 of the experiment–see S1 Appendix for a detailed description. Participants were directed to the tasks (hosted on www.millisecond.com) via the Prolific website, with task order counterbalanced across participants. Data from these experimental tasks have not been published elsewhere and were designed to test hypotheses presented in the current manuscript.

A 16-question post-experiment survey created using Qualtrics (https://www.qualtrics.com) included questions about demographics and how participants made decisions during the experimental task. Participants also completed the Autism-Spectrum Quotient (AQ), consisting of 50 questions used to evaluate traits associated with the autism spectrum in adults with normal intelligence (AQ) [17]. We included the AQ based on research demonstrating differences in performance in both go/no-go [18] and reversal learning tasks [19] among individuals with and without autism. The 66-question survey comprised Part 2 of the experiment. Survey questions and results are included with supplemental materials.

After completing both tasks through the Inquisit app, participants received a unique completion code to submit via the Prolific website. Within 24 hours of completing Part 1, we provided a link to Part 2. For up to three weeks, participants received reminders every 36 hours to complete Part 2 via the Prolific website.

We monitored attrition by counting the number of *dropouts*, defined as the number of participants that either (1) had not completed Part 2 after three weeks or (2) *returned* their submission, meaning that the participant decided to not to complete the study or withdrew their submission following completion (see https://researcher-help.prolific.co/hc/en-gb/articles/360009259434-Returned-submission-status).

**Group assignment.** Participants were randomly assigned to one of four groups. There were 40 individual-participant 'slots' per group. However, upon returning a submission, Prolific automatically made the vacant slot available to a new participant. Thus, *n* sizes were somewhat unequal across groups. All groups received the same set of instructions on their Prolific page–see S1 Appendix for details. Upon beginning Part 1, two groups (Groups Low Pay + Minimal Instructions and High Pay + Minimal Instructions) received the following instructions:

This is a two-part study. BE SURE TO COMPLETE BOTH PARTS. ***YOU WILL NOT RECEIVE PAYMENT UNTIL YOU HAVE COMPLETED PART 2.***

Note that this approach differs from Auer et al., who paid participants upon completion of each part of the study [12]. The two remaining groups (Groups Low Pay + Added Instructions and High Pay + Added Instructions) received the following *additional instructions*:

This is a two-part study consisting of (1) two short games and (2) a 66-question survey. BE SURE TO COMPLETE BOTH PARTS. ***YOU WILL NOT RECEIVE PAYMENT UNTIL YOU HAVE COMPLETED PART 2.*** Participants typically complete both parts in about 30 min. Some research participants find that the game is boring or repetitive and quit before it is finished. Others quit before completing the survey. If a sizable number of people quit before completing both tasks, the data quality will be compromised. However, our research depends on high-quality data. Thus, please make sure you do not mind completing the ENTIRE TASK before getting started.

Participants receiving these additional instructions could not proceed until they (1) typed in the statement "I will complete the entire task" and (2) provided their Prolific ID to consent to participate. Typing in the statement "I will complete the entire task" could be analogous to an honesty pledge in research on cheating [20]. Pledges might have different effects on behavior compared with other instructions described herein (prewarning, appeal to conscience). Nevertheless, we included the pledge to replicate methods used by Zhou and Fishbach [8]. Overall, these instructions included a prewarning (description of task), personalization (requirement to enter Prolific ID), and an appeal to conscience (instruction that dropping out could compromise data quality).

Low-pay groups (Low Pay + Minimal Instructions, $n = 64$, Low Pay + Added Instructions, $n = 58$) received pay advertised as \$7.25/hr. High-pay groups (High Pay + Minimal Instructions, $n = 51$; High Pay + Added Instructions, $n = 52$) received pay advertised as \$14.50/hr. Actual pay rates differed from the advertised pay rate due to variation in time required to complete the tasks among participants. The advertised pay rate was based on an estimated task duration of 20 min (Part 1) and 10 min (Part 2). Thus, participants in the high-pay groups earned \$4.84 (Part 1) and \$2.42 (Part 2) in base pay. Participants in the low-pay groups earned \$2.82 (Part 1) and \$1.21 (Part 2) in base pay. All participants were informed that points earned in Part 1 could result in additional payment (*bonus* payment). Participants were not informed of the bonus contingency, but bonuses were paid as \$0.001 per point (maximum of ~\$0.32). Participants received base pay (Part 1 and 2) and bonus payments within 36 hours of completing Part 2.

### Data analysis

Given that the advertised pay rate differed from the actual pay rate among participants, we first evaluated the relation between actual pay rate for Part 1 and total points earned during the two tasks. These data were not normally distributed; thus, we performed a Spearman's rank-order correlation. We conducted this analysis across all groups and within individual groups.

Next, we used a logistic regression analysis to ascertain the effects of pay (low pay, coded 0; high pay, coded 1) and the presence (coded 1) or absence of additional instructions (coded 0) on task completion (coded 1) or attrition (coded 0). More specifically, the goal of this analysis was to evaluate how pay, additional instructions, and the interaction between those variables influenced the odds of task completion. We performed this analysis in R [21] with the *glm* method included in the *stats* package.

### Results

Table 1 provides a summary of attrition rates. The Low Pay + Minimal Instructions group demonstrated the highest attrition rate (43.8%). Attrition was slightly lower when combining low pay with additional instructions (36.2%). Finally, attrition was lowest in the high-pay groups with or without additional instructions (~23%).

**Table 1. Attrition in each group.**

| Group | n (Incl. Dropouts) | Dropouts | | Attrition Rate |
|---|---|---|---|---|
| | | Part 1 | Part 2 | |
| Low Pay + Minimal Instructions | 64 | 24 | 4 | 43.75 |
| Low Pay + Added Instructions | 58 | 18 | 3 | 36.21 |
| High Pay + Minimal Instructions | 51 | 11 | 1 | 23.53 |
| High Pay + Added Instructions | 52 | 12 | 0 | 23.08 |

Actual completion times for both parts of the study were relatively consistent among groups, ranging from 27 to 30 min (Group High Pay + Added Instructions: $M$ = 29.2 min, $SD$ = 9.2; Group High Pay + Minimal Instructions: $M$ = 26.8 min, $SD$ = 5.2; Group Low Pay + Added Instructions: $M$ = 30.2 min, $SD$ = 9.2; Group Low Pay + Minimal Instructions: $M$ = 28.8 min, $SD$ = 6.1). Actual pay rates in all groups were higher than advertised pay rates but similar among High Pay groups (Group High Pay + Added Instructions: $M$ = $15.72, $SD$ = 2.8; Group High Pay + Minimal Instructions: $M$ = $16.74, $SD$ = 2.6) and Low Pay groups (Group Low Pay + Added Instructions: $M$ = $7.52, $SD$ = 1.7; Group Low Pay + Minimal Instructions: $M$ = $7.88, $SD$ = 1.5). Results of Spearman's rank-order correlation analysis indicated a weak (albeit statistically significant) positive correlation between actual pay rate for Part 1 and performance accuracy, defined as total points earned during the two tasks, $\rho(158)$ = .16, $p$ = .046; see Fig 1. This finding suggests that performance accuracy increases in simple reversal learning tasks with increases in pay rate. There were no significant correlations between these variables within individual groups (data not shown).

Finally, Table 2 shows the results of the logistic regression analysis. Only one factor (pay) significantly affected the odds of task completion. More specifically, an advertised base pay of

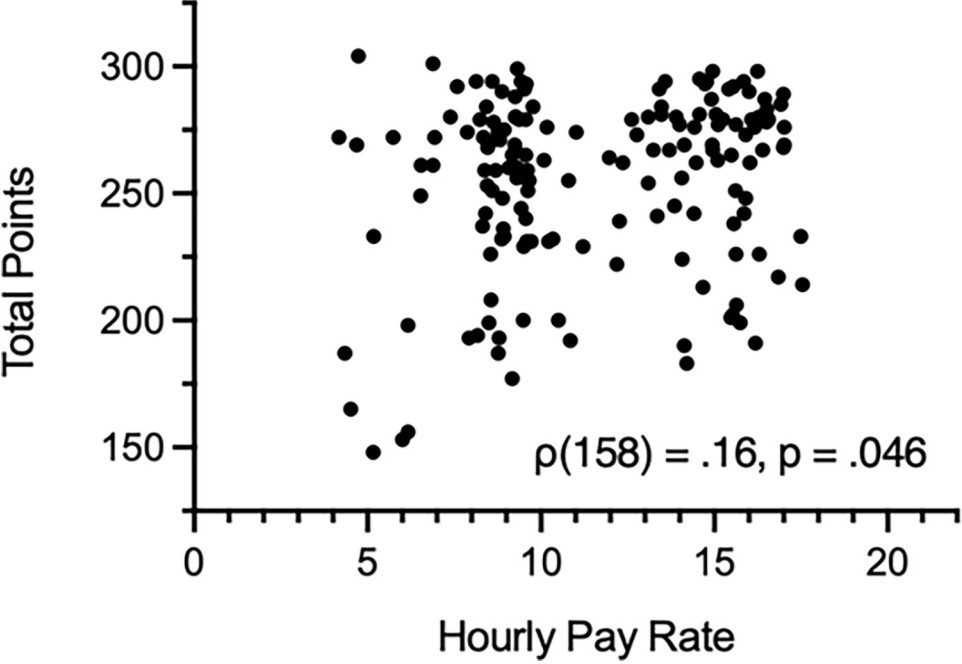

**Fig 1. Correlation between pay rate and total points.** *Note.* Perfectly accurate performance would result in ~324 points.

**Table 2. Results of logistic regression.**

| Factor | β (SE) | Z | p |
|---|---|---|---|
| Intercept | 0.25 (0.25) | 1.00 | .319 |
| High Pay | 0.93 (0.42) | 2.23* | .026 |
| Added Instructions | 0.32 (0.37) | 0.85 | .397 |
| High Pay*Added Instructions | -0.29 (0.60) | -0.49 | .627 |

$14.50/hr increased odds of task completion by a factor of 2.53 (95% CI [1.11, 5.77] compared to advertised base pay of $7.25/hr.

## Discussion

We examined approaches to address both attrition and data quality when recruiting participants via crowdsourcing. Although there are many possible measures of data quality (e.g., number of characters on survey responses), performance accuracy on a simple operant task served as the measure of data quality in the present study. Instead of using MTurk, we recruited participants using Prolific, a crowdsourcing platform specifically catering to researchers. Our primary findings were that higher pay corresponded with reduced attrition and greater performance accuracy. Additional instructions did not affect these outcomes.

Additional instructions–that is, informing participants about the task, asking for personal information on the consent form, and describing the negative impact of attrition on research quality–did not impact attrition relative to minimal instructions (cf. [8]). There are several possible reasons for this null effect. Whereas prior research evaluated whether additional instructions could mitigate attrition during a 5-min survey at a pay rate of $6/hr [8], we arranged an operant behavioral task and survey that took approximately 30 min to complete, which is likely more representative of the type of task and duration typically arranged in experimental psychological research. We paid participants a minimum of $7.25/hr and up to double this rate. As a result, it is possible that additional instructions effectively reduce attrition only (1) when tasks are brief and/or (2) within a lower range of pay rates. For example, we saw a slight (but not statistically significant) reduction in attrition across low-pay groups earning $7.25/hr and experiencing the presence or absence of additional instructions (from 44% to 36%). However, attrition was similar in high-pay groups earning $14.50/hr regardless of instructions (~23%). Future research could examine effects of additional instructions across a range of task types, durations, and payment amounts.

Consistent with findings with participants recruited from MTurk [7, 12], results of the present study also suggested that increasing advertised pay rates from $7.25 to $14.50/hr could reduce attrition with Prolific participants. We extended prior research by using what might be considered a relatively high minimum-pay rate consistent with the US minimum wage (cf. [12]). In contrast, Auer et al. evaluated effects of hourly pay ranging between $1.11 and $10.27 on attrition, with most participants (89%) earning $5.75/hr or less [12]. These low pay rates reflect an important difference between MTurk and Prolific; that is, MTurk does not regulate participant payment. When recruiting participants via Prolific, it was not possible to examine effects of the lower pay rates used by Auer et al.; researchers were required to pay participants no less than $6.50/hr at the time that we collected our data in 2021. This has since been increased to $8/hr, as of June 2022. Nevertheless, our findings suggest that the mitigating effects of increased pay on attrition generalize to higher overall pay rates.

A related finding was that higher actual pay rates were associated with more accurate performance on common tasks used to assess inhibitory control and behavioral flexibility [22; cf.

23]. As with studies examining effects of pay on attrition, MTurk studies examining effects of pay on data quality have paid participants at low rates–i.e., below the US minimum wage of $7.25. For example, Buhrmester et al. examined effects of hourly pay ranging between $0.04 and $6 on alpha reliabilities in personality questionnaires as a measure of data quality, with nearly three-quarters of participants earning $1.20/hr or less [23]. While Buhrmester et al. demonstrated no effect of pay on data quality among US and non-US participants, our findings could suggest that this null effect was due to (1) the lower overall range of pay rates examined (but see [24]) or (2) differences among US and non-US participants [22, 25]. Related to the second point, previous research has shown that increasing hourly pay to $10 from a minimum rate of $4 [22] or $0.20 [25] results in higher-quality data among participants based in India but not in the US. The present study is the first to demonstrate that pay could also impact data quality among US-based participants when increasing hourly pay from $7.25 to double that rate.

One limitation of the present study is that demographic variables (e.g., age, sex, income) were not controlled for or assessed when examining performance accuracy. There is some evidence for age- and sex-related differences in performance on tasks assessing inhibitory control. For example, studies have shown that, among healthy participants completing a go/no-go task, older adults make fewer no-go errors compared to younger adults [26] and that women demonstrate greater inhibition on no-go trials compared to men [27]. In the present study, the demographic survey comprised second part of the study, and approximately one-third of participants dropped out before providing demographic information. We presented the demographic survey after the experimental tasks to mirror the presentation of tasks in previous crowdsourcing research [28, 29]. Nevertheless, it is possible that demographic variables and performance accuracy interact in ways that are not accounted for in the present study. Future researchers should evaluate how demographic variables influence the relation between pay rate and performance accuracy.

A second limitation of the present study is that we used a broad definition of attrition. That is, attrition included both failure to (1) complete an ongoing experimental task, and (2) return to the Prolific website to complete a second task (survey). Some researchers might be interested in addressing within-task attrition specifically, given that is rarely tracked or reported [7] and has raised significant validity concerns [8]. Thus, future researchers should consider parsing out the effect of independent variables such as pay rate on each form of attrition. Distinguishing between these forms of attrition would further improve our understanding of how manipulations such as increased pay could improve the quality of crowdsourcing research.

A third limitation is that participation in the present study required installing the Inquisit Player (https://www.millisecond.com/products/inquisitplayer). This software likely does not present issues with compatibility: it can be quickly and easily installed on Windows and Mac devices, as well as Android devices and Chromebooks. However, it is still possible that some participants did not complete the study because either they could not or did not want to install the software. Therefore, future studies might consider experimental tasks that are hosted on the web and do not require software installation [28].

Altogether, our findings demonstrated that paying participants at a rate well beyond the US minimum wage, and closer to a *living wage* (see livingwage.mit.edu/), not only reduces attrition but also is correlated with higher-quality performance. Low pay rates are commonplace on MTurk, with the average US-based MTurk worker earning a median wage of $3.01/hr, significantly below the minimum wage [30]. This is despite research showing that monetary compensation is the highest-rated motivation for study completion among US-based MTurk participants [25]. Findings from the present study build on previous research and suggest that fair compensation mutually benefits participants and researchers. For example, other research

has shown that fair compensation could facilitate recruitment and reduce attrition in groups traditionally underrepresented in research, such as individuals of lower socio-economic status, who cannot afford to dedicate time to research in the absence of adequate payment [31]. The present findings further suggest that paying participants closer to a living wage not only mitigates threats to internal validity via reduced attrition but also could facilitate collection of higher-quality data. Beyond these benefits, researchers should strongly consider their ethical obligation to compensate crowdsourcing participants fairly, even when minimum payment requirements are not imposed (e.g., MTurk).

## Supporting information

**S1 Appendix. Additional task and participant information.**
(DOCX)

## Author Contributions

**Conceptualization:** Carolyn M. Ritchey, Corina Jimenez-Gomez, Christopher A. Podlesnik.

**Data curation:** Carolyn M. Ritchey.

**Formal analysis:** Carolyn M. Ritchey.

**Funding acquisition:** Christopher A. Podlesnik.

**Methodology:** Carolyn M. Ritchey.

**Supervision:** Corina Jimenez-Gomez, Christopher A. Podlesnik.

**Visualization:** Carolyn M. Ritchey.

**Writing – original draft:** Carolyn M. Ritchey.

**Writing – review & editing:** Corina Jimenez-Gomez, Christopher A. Podlesnik.

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
