## [Decision Letter · Decision Letter 0]

22 Jun 2023

PONE-D-23-00174Effects of Pay Rate and Instructions on Attrition in Crowdsourcing Research

PLOS ONE

Dear Dr. Ritchey,

Thank you for submitting your manuscript to PLOS ONE. After careful consideration, we feel that it has merit but does not fully meet PLOS ONE’s publication criteria as it currently stands. Therefore, we invite you to submit a revised version of the manuscript that addresses the points raised during the review process.

In the revised version, please address all the comments provided by the reviewers. Pay particular attention to reviewer 2 and provide any information regarding the qualifications of the Prolific participants, as well as some form of systematic screening of the qualitative responses to rule out the potential for any inattentive data. These are detailed in the reviewer comments.

We look forward to receiving your revised manuscript.

Kind regards,

Jacinto Estima

Academic Editor

PLOS ONE

Reviewers' comments:

Reviewer's Responses to Questions

**Comments to the Author**

1. Is the manuscript technically sound, and do the data support the conclusions?

Reviewer #1: Yes

Reviewer #2: Partly

Reviewer #3: Yes

2. Has the statistical analysis been performed appropriately and rigorously? 

Reviewer #1: Yes

Reviewer #2: No

Reviewer #3: Yes

3. Have the authors made all data underlying the findings in their manuscript fully available?

Reviewer #1: Yes

Reviewer #2: Yes

Reviewer #3: Yes

4. Is the manuscript presented in an intelligible fashion and written in standard English?

Reviewer #1: Yes

Reviewer #2: Yes

Reviewer #3: Yes

5. Review Comments to the Author

Reviewer #1: Given the increased reliance on the convenience and cost-effectiveness of rapidly recruiting large sample sizes using commercial market crowdsourcing agents, the manuscript addresses an important and highly relevant issue in psychology. The primary aim is to evaluate the impact of differing levels of payment and instructions on attrition in one market research sourced (Prolific) sample. The findings are important is offering strategies to reduce attrition rates using appropriate pay scales, and the minimal effects of additional information appealing to the research need for participants to complete the surveys/tasks.

The title is clear and concise in accurately setting out the study’s topic. The Abstract is clear but could benefit from the inclusion of the actual sample size employed in the study.

Overall, the manuscript is well written, clear in its objectives regarding factors affecting attrition, and the measures and statistics are appropriate. The limitations are noted.

There are several points that is offered for consideration by the authors. These are designed to enhance the overall quality and clarity of some aspects of its content.

The first sentence of the manuscript is unclear as to whether the authors are asserting that half of all participants for research in psychology and social sciences are recruited through crowdsourcing, or if half of all research studies in psychology and social sciences use crowdsourcing to recruit participants.

Related to this, is there any time frame as to the assertion underlying this claim given that crowdsourcing is relatively recent compared to studies undertaken in psychology and social sciences since the turn of the last century? Clearly there is a trend to use crowdsourcing but when did this generally commence?

On page 9 and elsewhere, it is claimed that the findings suggest that performance accuracy increases in simple reversal learning tasks with increased pay rates, and that data quality was assessed. There is a need to include more information regarding the definition and criteria used by the authors to assess both accuracy and data quality. Was it simply based on obtained scores on these tasks, and could other factors be excluded from influencing such performance (e.g., subsample characteristics given we do not have data on age and sex distribution)?

On page 11, ‘…hourly pay with ranging between…” should read ‘…hourly pay with ranging between…”.

Regarding the conclusion section, it would be interesting (optional) for the authors to comment on the implications of their findings linking higher pay rates to lower attrition for research costs. In their introduction, the authors rightly note that one benefit to researchers for using crowdsourced samples is the relatively low cost. This benefit is potentially compromised given the study’s findings. For example, using pay in the vicinity of under $1.00 to around $7 per participant is cost-effective but once the rates increase to around $15-17, the cost to researchers becomes quite considerable once samples reach n=1,000+. Do researchers need to sacrifice attrition in preference for low costs in participant recruitment and is this a scientifically justified course of action? The question of ethics of paying miniscule pay rates to participants for their time is another issue for consideration.

Reviewer #2: I appreciated the opportunity to review Ritchey and colleagues’ manuscript, “Effects of Pay Rate and Instructions on Attrition in Crowdsourcing Research,” submitted for consideration as a Research Article in PLOS ONE. As a crowdsourcing researcher myself, I found the topic to be interesting and important. Indeed, much of the work on the structure and logistics of crowdsourcing research was collected in the early days of Amazon Mechanical Turk. Much has changed in the crowdsourcing research landscape, particularly with the emergence of new competitor platforms such as Prolific (used in the present study).

Despite my enthusiasm for this topic, there are some procedural issues that limit the contributions of this work—however, these *might* be resolvable in revision. I thereby recommend: Major Revision.

What follows are the major concerns guiding my recommendation, along with recommendations where relevant.

1. The authors do not provide any information regarding the qualifications of the Prolific participants. When constructing a Prolific study, researchers are permitted the chance to “pre-screen” potential participants. A common approach is to limit recruitment to participants from certain countries and who have completed a minimum number of surveys already (e.g., 100) with a minimum approval rating. Perhaps the researchers used such screenings but failed to report them? If no prescreening was employed, a strong argument should be made as to why they felt these were unnecessary. From personal experience, data are substantially improved (along with attrition) when recruiting participants with established experience in the platform.

2. In addition to minimum qualifications, it is common practice to use some form of quality control checking to ensure data are indeed valid. One tactic we have used in our research is to code qualitative data, of which the researchers have several potential questions. In my casual review, there are a sizable number of participants who did not answer any qualitative questions, which raises some skepticism as to the quality of their responses. I would advise the researchers to provide some form of systematic screening of the qualitative responses to rule out the potential for any inattentive data. It appears the researchers informed participants to “Leave the question blank if you prefer not to answer,” which complicates such an analysis. Regardless, I advise some screening process of some kind.

3. Related to point #3 above, there seems to be many missed opportunities for additional analyses. For example, I ran a quick qualitative analysis on the data and found an interesting pattern in responses to the question: “What do you think was the overall purpose of the study you just completed?” On the average, high pay participants responded with about 30 total characters, while low pay participants responded with about 20 total characters. However, this was not statistically significant. I also found that high pay participants were less likely to put an “I don’t know” response—but again, this isn’t statistically significant.

4. The question “What gender/sex do you identify with?” contains only 3 different responses: Male, Female, or Other. Restricting participants to essentially a binary choice does not seem like an inclusive survey, which could impact participant responses to other questions, or perhaps even their attrition.

5. Regarding attrition, it seems possible that some participants may not have completed the study because it required installing the Inquisit plug-in. It is thereby possible that some variance could be explained by this artifact. To better isolate contributing variables, future studies might consider keeping all tasks within one survey and avoiding the use of plug-in installs.

6. Considering this study included pay amount as a primary IV, it is curious that the researchers did not include an income question in their demographics.

7. It appears that Prolific IDs may be present in the spreadsheet. I advise against sharing this information.

In sum, this was an interesting research question, and I am glad to see empirical research in this domain. However, the rigor of the crowdsourcing methods is not on par with the bulk of crowdsourcing studies I read—which is problematic given this is a study to potentially inform standards of crowdsourcing methods. The missed opportunities for additional analyses—whether due to incomplete demographic details or qualitative responses left unexplored—renders the analyses less rigorous than what is expected for publication in PLOS ONE. I hope the researchers will continue this work and in doing so make the improvements suggested above. I recommend Major Revision given that I am unsure whether the researchers have the data or information available to address my concerns. If so, a revision may be acceptable. If these data do not exist, I do not recommend publication.

Reviewer #3: The authors’ investigated the effects of pay rate and additional instructions on study attrition on the Prolific crowdsourcing platform via a two part study. In study 1, the investigators recruited participants into four groups (low pay/minimal instruction; low pay/added instruction; high pay/minimal instruction; high pay/added instruction) and engaged participants in a number of psychological tasks. Study 2 consisted of questionnaires regarding demographics, experimental task performance, and Autism Spectrum. Participants were reminded periodically over a 3 week period to complete study 2. The investigators reported that higher pay was associated with lower attrition, regardless of instructional category.

I commend the authors for the simple experimental question and design and statistical methods/results that support the conclusions. I think this manuscript could make a nice contribution to the burgeoning field of online crowdsourced psychological research. Overall I found the manuscript to be well written and can recommend publication in the current form.

6. PLOS authors have the option to publish the peer review history of their article (what does this mean?). If published, this will include your full peer review and any attached files.

Reviewer #1: No

Reviewer #2: No

Reviewer #3: No

---

## [Author Response · Author response to Decision Letter 0]

6 Aug 2023

The Abstract is clear but could benefit from the inclusion of the actual sample size employed in the study.

We added this information to the abstract.

The first sentence of the manuscript is unclear as to whether the authors are asserting that half of all participants for research in psychology and social sciences are recruited through crowdsourcing, or if half of all research studies in psychology and social sciences use crowdsourcing to recruit participants.

The latter is correct. We have changed the first sentence to improve clarity.

Related to this, is there any time frame as to the assertion underlying this claim given that crowdsourcing is relatively recent compared to studies undertaken in psychology and social sciences since the turn of the last century? Clearly there is a trend to use crowdsourcing but when did this generally commence?

This began a little over a decade ago, we added this information to the first sentence (e.g., http://journal.sjdm.org/10/10630a/jdm10630a.pdf)

On page 9 and elsewhere, it is claimed that the findings suggest that performance accuracy increases in simple reversal learning tasks with increased pay rates, and that data quality was assessed. There is a need to include more information regarding the definition and criteria used by the authors to assess both accuracy and data quality. Was it simply based on obtained scores on these tasks, and could other factors be excluded from influencing such performance (e.g., subsample characteristics given we do not have data on age and sex distribution)?

We added definitions for both. Regarding the last point, we mention on page 12 that one limitation is that demographic variables (e.g., age, sex) were not controlled for or assessed when examining performance accuracy.

On page 11, ‘…hourly pay with ranging between…” should read ‘…hourly pay with ranging between…”.

We changed this to “hourly pay ranging between…”

The authors do not provide any information regarding the qualifications of the Prolific participants. When constructing a Prolific study, researchers are permitted the chance to “pre-screen” potential participants. A common approach is to limit recruitment to participants from certain countries and who have completed a minimum number of surveys already (e.g., 100) with a minimum approval rating. Perhaps the researchers used such screenings but failed to report them? If no prescreening was employed, a strong argument should be made as to why they felt these were unnecessary. From personal experience, data are substantially improved (along with attrition) when recruiting participants with established experience in the platform.

We did not use prescreening because research suggests that, unlike MTurk, Prolific generally provides high-quality data without prescreening (Peer et al., 2021). We now include this information in the Methods section.

In addition to minimum qualifications, it is common practice to use some form of quality control checking to ensure data are indeed valid. One tactic we have used in our research is to code qualitative data, of which the researchers have several potential questions. In my casual review, there are a sizable number of participants who did not answer any qualitative questions, which raises some skepticism as to the quality of their responses. I would advise the researchers to provide some form of systematic screening of the qualitative responses to rule out the potential for any inattentive data. It appears the researchers informed participants to “Leave the question blank if you prefer not to answer,” which complicates such an analysis. Regardless, I advise some screening process of some kind.

We did thoroughly screen qualitative data – for example, we looked for nonsensical response patterns or responses that were not in English. However, we did not exclude any data sets for this reason. We added a note about this process to supplemental survey data. Unlike MTurk, Prolific takes a number of steps to eliminate bots (see https://www.prolific.co/blog/bots-and-data-quality-on-crowdsourcing-platforms).

Related to point #3 above, there seems to be many missed opportunities for additional analyses. For example, I ran a quick qualitative analysis on the data and found an interesting pattern in responses to the question: “What do you think was the overall purpose of the study you just completed?” On the average, high pay participants responded with about 30 total characters, while low pay participants responded with about 20 total characters. However, this was not statistically significant. I also found that high pay participants were less likely to put an “I don’t know” response—but again, this isn’t statistically significant.

We appreciate this point and agree that there are many possible measures of data quality. We added a footnote with the suggestion to use number of characters on specific survey questions as a measure of data quality – see page 10.

The question “What gender/sex do you identify with?” contains only 3 different responses: Male, Female, or Other. Restricting participants to essentially a binary choice does not seem like an inclusive survey, which could impact participant responses to other questions, or perhaps even their attrition.

Thank you for this important suggestion. We agree and will modify this questionnaire for future research.

Regarding attrition, it seems possible that some participants may not have completed the study because it required installing the Inquisit plug-in. It is thereby possible that some variance could be explained by this artifact. To better isolate contributing variables, future studies might consider keeping all tasks within one survey and avoiding the use of plug-in installs.

We also appreciate this point and have added this as a limitation in the Discussion.

Considering this study included pay amount as a primary IV, it is curious that the researchers did not include an income question in their demographics.

We suggest that future researchers should evaluate how demographic variables influence the relation between pay rate and performance accuracy on Page 12, and we now mention income as another demographic variable that could be evaluated in that section.

It appears that Prolific IDs may be present in the spreadsheet. I advise against sharing this information.

These have been removed.

---

## [Decision Letter · Decision Letter 1]

20 Sep 2023

Effects of Pay Rate and Instructions on Attrition in Crowdsourcing Research

PONE-D-23-00174R1

Dear Dr. Ritchey,

We’re pleased to inform you that your manuscript has been judged scientifically suitable for publication and will be formally accepted for publication once it meets all outstanding technical requirements.

Kind regards,

Jacinto Estima

Academic Editor

PLOS ONE

Additional Editor Comments (optional):

Thank you very much for addressing all the comments provided by the reviewers.

Reviewers' comments:

Reviewer's Responses to Questions

**Comments to the Author**

1. If the authors have adequately addressed your comments raised in a previous round of review and you feel that this manuscript is now acceptable for publication, you may indicate that here to bypass the “Comments to the Author” section, enter your conflict of interest statement in the “Confidential to Editor” section, and submit your "Accept" recommendation.

Reviewer #1: All comments have been addressed

Reviewer #3: All comments have been addressed

2. Is the manuscript technically sound, and do the data support the conclusions?

Reviewer #1: Yes

Reviewer #3: Yes

3. Has the statistical analysis been performed appropriately and rigorously? 

Reviewer #1: Yes

Reviewer #3: Yes

4. Have the authors made all data underlying the findings in their manuscript fully available?

Reviewer #1: Yes

Reviewer #3: Yes

5. Is the manuscript presented in an intelligible fashion and written in standard English?

Reviewer #1: Yes

Reviewer #3: Yes

6. Review Comments to the Author

Reviewer #1: Thank you for attending to the original comments. I have no further comments to offer. All issues have been addressed to satisfaction.

Reviewer #3: Given my prior recommendation for publication, as well as the authors' responsiveness to other reviewer comments, I am pleased to recommend this manuscript again for publication.

7. PLOS authors have the option to publish the peer review history of their article (what does this mean?). If published, this will include your full peer review and any attached files.

Reviewer #1: No

Reviewer #3: No

---

## [Editor Report · Acceptance letter]

26 Sep 2023

PONE-D-23-00174R1 

Effects of Pay Rate and Instructions on Attrition in Crowdsourcing Research 

Dear Dr. Ritchey:

I'm pleased to inform you that your manuscript has been deemed suitable for publication in PLOS ONE. Congratulations! Your manuscript is now with our production department. 

Kind regards, 

on behalf of

Dr. Jacinto Estima 

Academic Editor

PLOS ONE